# A Review of the Clinical Implications of Cachexia, Sarcopenia, and BMI in Patients with Peritoneal Carcinomatosis Receiving Cytoreductive Surgery and Hyperthermic Intraperitoneal Chemotherapy

**DOI:** 10.3390/cancers14122853

**Published:** 2022-06-09

**Authors:** Devon C. Freudenberger, Vignesh Vudatha, Andrea N. Riner, Kelly M. Herremans, Leopoldo J. Fernandez, Jose G. Trevino

**Affiliations:** 1Department of Surgery, Virginia Commonwealth University School of Medicine, 1200 E Broad St., P.O. Box 980011, Richmond, VA 23219, USA; devon.freudenberger@vcuhealth.org (D.C.F.); vignesh.vudatha@vcuhealth.org (V.V.); leopoldo.fernandez@vcuhealth.org (L.J.F.); 2Department of Surgery, University of Florida College of Medicine, 1600 SW Archer Rd., P.O. Box 100287, Gainesville, FL 32610, USA; andrea.riner@surgery.ufl.edu (A.N.R.); kelly.herremans@surgery.ufl.edu (K.M.H.)

**Keywords:** peritoneal carcinomatosis, cytoreductive surgery, HIPEC (hyperthermic intraperitoneal chemotherapy), cachexia, sarcopenia, BMI

## Abstract

**Simple Summary:**

Patients with peritoneal carcinomatosis from varying cancers may be affected by weight loss and decreased muscle mass, the hallmarks of cachexia. These patients can undergo surgical management via cytoreductive surgery and hyperthermic intraperitoneal chemotherapy to improve their overall survival. Here, we review the current literature investigating the impact of sarcopenia, cachexia, and body mass index on outcomes in a patient population that undergo surgical treatment. The results vary across the studies suggesting that further investigation is necessary to better understand the impact of these entities on postoperative outcomes and survival.

**Abstract:**

Peritoneal carcinomatosis (PC) is the dissemination of cancer throughout the peritoneal cavity. Cytoreductive surgery (CRS) and hyperthermic intraperitoneal chemotherapy (HIPEC) is the surgical treatment of choice in highly selected patients. The aim of this narrative review was to assess the impact of cachexia, sarcopenia, and body mass index (BMI) on patient outcomes for patients undergoing CRS and HIPEC for peritoneal carcinomatosis. A narrative review was performed and articles pertaining to cachexia, sarcopenia, BMI, peritoneal carcinomatosis, and CRS/HIPEC were reviewed and selected. In total, 3041 articles were screened and seven original studies met the inclusion criteria. In summary, obesity was found to not be a contraindication to surgery, but the impact of BMI was variable across the spectrum. Decreased skeletal muscle mass was found to be associated with poorer postoperative outcomes in three studies and with worse overall survival in two. With limited data, evaluating the impact of BMI, sarcopenia, and cachexia on patients with PC undergoing CRS and HIPEC was difficult as most studies included heterogeneous cancer patient populations; thus, postoperative outcomes and survival were inconsistent across studies. More research is needed to better understand its impact and to better generalize the results for each cancer subset treated with CRS and HIPEC across diverse patient populations.

## 1. Introduction

Peritoneal carcinomatosis (PC), or metastasis, refers to the dissemination of primary and recurrent gastrointestinal, gynecologic malignancy (e.g., ovarian, appendiceal, colorectal, and gastric), or peritoneal mesothelioma in the peritoneal cavity [1]. The presence of PC portends a poor prognosis with the majority of patients having a life expectancy of less than one year without treatment [2,3]. In select patients, treatment consists of cytoreductive surgery (CRS) and hyperthermic intraperitoneal chemotherapy (HIPEC), which involves surgical debulking to a tumor size of less than 2 mm followed immediately by intraperitoneal administration of heated chemotherapy, with agent and instillation time dictated by the type of cancer [4]. With the advent of CRS and HIPEC, 5-year overall survival has increased from less than 10% to upwards of 40–50% for PC patients [5,6,7].

Roughly 66% of patients with PC are diagnosed with cachexia and experience the adverse effects that accompany the syndrome [8]. Cancer cachexia is a multifactorial, paraneoplastic syndrome characterized by negative protein and energy balance, heightened inflammatory state, and loss of skeletal muscle mass that cannot be reversed with conventional nutritional support [9,10,11,12,13,14]. In the clinical setting, cachexia presents as involuntary weight loss, lack of appetite, and progressive physical impairment [9,10,11]. Roughly half of all cancer patients develop cachexia, with certain malignancies having higher incidence, such as pancreatic and gastric cancer. It can impair immunity, reduce patient tolerance of chemotherapy, cause respiratory muscle impairment leading to cardiopulmonary failure, and promote hepatic dysfunction thus contributing to the patient’s nutritional deficit [9,10,11]. Cancer patients also typically require high dose opioid regimens for pain control, which can further exacerbate a patient’s nutritional status via decreased gastrointestinal motility. Diagnosis of cachexia in the preoperative setting is associated with poor postoperative outcomes and increased surgical complications [7,12]. This cycle is further perpetuated by chemotherapy and radiotherapy, which can exacerbate these symptoms via induction of pro-cachectic factors such as nuclear factor-kappa B (NF-κB) [11,13]. Overall, cachectic cancer patients have a worse prognosis, with cachexia being responsible for 20% of cancer-related deaths [10].

The image of a cachectic patient is typically portrayed as someone who is thin and emaciated. With the ongoing public health epidemic of obesity, sarcopenic obesity is often seen as reality and can have similar clinical outcomes to the typically portrayed cachectic patient. A high body mass index (BMI) has the potential to mask the underlying skeletal muscle loss that a cancer patient may have, lowering the suspicion for cachexia in this patient population. The presence and effects of cachexia have been shown across the spectrum of BMIs (i.e., underweight to morbid obesity), and as such BMI cannot always be used to assess a patient’s degree of cachexia [14,15]. The impact of BMI itself on cancer patient outcomes has also become an area of interest as research has delved into the “obesity paradox” where obesity has been shown at times to be a protective factor in cancer [16].

Cancer cachexia, its mechanisms and impact on patient outcomes, and patient BMI have been studied extensively, particularly in extraperitoneal malignancies such as pancreatic, lung, and head and neck cancers [7,9,10,11,17,18,19]. One cohort that has not been substantially explored is patients with diffuse isolated peritoneal disease (peritoneal carcinomatosis). The aim of this paper is to review the current literature regarding the mechanisms of cachexia in the PC patients who undergo CRS and HIPEC, and the effects of skeletal muscle loss or sarcopenia, cachexia, and BMI on patient outcomes. Understanding the effects of these on PC patients treated with CRS and HIPEC may allow further improvements in preoperative patient assessment, selection, optimization, and management in the perioperative setting to improve overall outcomes and ultimately survival.

## 2. Materials and Methods

A literature search was performed to identify original research pertaining to sarcopenia, cachexia, BMI, and peritoneal carcinomatosis. An advanced search was performed on both the PubMed and Ex Libris Discovery Search Engines from 1960 through to 24 August 2021. The following search string was utilized: ((Carcinomatosis) AND (cachexia OR sarcopenia)) OR ((HIPEC) AND (cachexia OR sarcopenia)) OR ((peritoneal metastasis) AND (cachexia OR sarcopenia)). Studies were limited to peer-reviewed journal articles and those published in the English language. Studies were excluded if they were abstract only, not published in full, or were duplicate articles. 

Studies were eligible for inclusion if they addressed the impact of cachexia in patients with PC treated with CRS and HIPEC. This included studies that explored the impact of BMI, preoperative nutritional markers, and sarcopenia on patients with PC of any origin. Articles were excluded if the patients did not receive treatment in the form of cytoreductive surgery and HIPEC. Studies were also excluded if they addressed patients with carcinoma or carcinosarcoma but no peritoneal metastasis. 

Two individuals (D.C.F. and V.V.) independently screened titles and abstracts based on the above search terms and inclusion/exclusion criteria. All articles that met inclusion criteria were independently reviewed by both reviewers in their entirety. 

### Cachexia and Sarcopenia

Of note, sarcopenia was included in the final search because after performing a preliminary review, as it was noted that this term was used interchangeably with cachexia in the literature. Although both concepts have a basis in muscle loss, they are two separate clinical entities with two distinct sets of consensus definitions. Per international consensus in 2011, a diagnosis of cancer cachexia can be made if the patient meets one of the following criteria: weight loss > 5% over 6 months, BMI < 20 kg/m^2^ and any weight loss > 2%, or appendicular skeletal muscle index indicative of sarcopenia and any weight loss > 2% [20]. Sarcopenia is a component of cachexia that is characterized by progressive loss of skeletal muscle mass and strength. While prior guidelines including the aforementioned cachexia definition focused on skeletal muscle quantity, the most recent 2018 consensus by the European Working Group on Sarcopenia in Older People (EWGSOP) has shifted towards prioritizing muscle strength. Probable sarcopenia is identified based on low muscle strength with confirmation through documentation of low muscle quantity and physical performance [21,22]. Both terms were included in our literature search to ensure all relevant studies were identified.

## 3. Results

The resulting search on both databases yielded 3041 articles including duplicates (Figure 1). Based on the inclusion and exclusion criteria, nine original studies were identified that focused on cachexia, sarcopenia, BMI, and PC, and met initial evaluation for full review. Upon full review, two articles were excluded as the patient population did not undergo CRS and HIPEC. Of the remaining seven studies examining patients that underwent CRS and HIPEC, five studies evaluated the impact of sarcopenia/cachexia on patient outcomes and two studies evaluated the impact of BMI on patient outcomes (Table 1).

Five of the seven studies (71.4%) were completed in Europe (France, Netherlands, and Turkey) with the remaining two (28.6%) completed in the United States. Three of the seven studies (42.9%) exclusively evaluated patients with primary colorectal cancer, whereas the remaining four (57.1%) included multiple primary cancers, including colorectal, appendiceal, ovarian, and mesothelioma. Five of the seven studies specifically analyzed the impact of sarcopenia on overall survival and post-operative complications. The definition or criteria for sarcopenia varied across the studies. 

## 4. Discussion

### 4.1. Mechanisms of Cachexia and Peritoneal Metastasis

The main impetus for cancer cachexia is the tumor–host interaction and the resulting chronic systemic inflammation. Tumor cells, as well as host immune cells, contribute to this inflammatory state by releasing pro-cachectic cytokines such as TNFα, IL-1, IL-6, and IFN-γ [9,17,18,30]. These proinflammatory molecules play a direct role in muscle degradation and anorexia. TNFα, also known as cachectin, drives skeletal muscle catabolism by inducing ubiquitin-mediated proteasome degradation (UPR) via the NF-κB pathway [9,30,31]. Autophagy and calpain proteases have also been implicated in the degradative process by directly promoting skeletal muscle proteolysis [11,30]. Similarly, cardiac muscle fibers undergo UPR via signaling from TNFα and IL-1 [11,30]. TNF-α also synergizes with IL-1 and IFN-γ to negatively impact appetite. They cross the blood–brain barrier and induce a series of neurohormonal alterations to promote anorexia and muscle wasting. They increase levels of available serotonin, reduce secretion of appetite stimulating hormones such as neuropeptide Y and ghrelin, and trigger the HPA axis, thus promoting skeletal muscle and adipose tissue breakdown [9,11,30]. 

In the context of PC, both cachexia and peritoneal spread are inextricably linked and promote each other. Studies in pancreatic cancer models have demonstrated that inflammatory moieties such as TGF-β and IL-6 are implicated in the process of metastasis [32]. Components of the tumor immune environment, particularly tumor-associated macrophages (TAMs), play a significant role in neoplastic growth and metastasis. TAMs produce many different angiogenic and lymphangiogenic growth factors, such as VEGF, which promotes tumor progression and loco-regional spread [33]. These cells, along with TNFα and IL-1, also induce the expression of adhesion molecules and chemotactic cytokines in mesothelial cells. At the same time, intercellular adhesion molecules on the primary tumor, such as E-cadherin, are downregulated, thus promoting peritoneal spread [34,35,36]. Peritoneal metastatic nodules can also migrate to the omentum via chemokine homing from omental adipokines and tumor PGK-1 expression. Omental adipocytes then release triglycerides, which undergo lipolysis. The resulting free fatty acids are used by the omental cancer cells for growth and further spread. Once peritoneal seeding has occurred, these new implants can further exacerbate the patient’s cachectic state. This occurs not only through the cellular mechanisms mentioned above, but also at a tissue and organ level. The new lesions can infiltrate the bowel wall, leading to mechanical obstruction and decreased gastrointestinal motility [8]. PC also leads to malignant ascites, which can also worsen these obstructions. The repeated drainage of the ascites can help relieve abdominal fullness, but it will subsequently lead to protein deprivation and renal dysfunction [8]. Lastly, the presence of PC will likely necessitate chemotherapy and an opioid-based pain regimen, which promotes nausea and anorexia [8,37,38]. The combination of decreased caloric intake from gastrointestinal symptoms and increased metabolism from chronic inflammation induced by the peritoneal metastasis further exacerbate the patient’s cachectic state. 

### 4.2. Clinical Impact of Obesity and Body Mass Index

Obesity is a major public health concern today. Obesity, as defined by BMI greater than 30 kg/m^2^, is associated with many physiologic and inflammatory changes that increase individuals’ risks for major medical comorbidities, including Type II diabetes mellitus, coronary artery disease, hypertension, obstructive sleep apnea, obesity hypoventilation syndrome, pulmonary hypertension, non-alcoholic fatty liver disease, venous thromboembolism, and nutritional deficiencies. Importantly, obesity is also associated with increased risk of cancer [39]. 

Given these comorbidities, obese patients can be expected to have more negative surgical outcomes in the perioperative setting. In a 2008 retrospective study of data from the American College of Surgeons National Surgical Quality Improvement Program (NSQIP), the authors investigated the impact of obesity on perioperative outcomes for 2258 patients who underwent major intra-abdominal cancer surgery (esophagectomy, gastrectomy, hepatectomy, pancreatectomy, and low-anterior resection/proctocolectomy). Interestingly, they showed that obesity was not a risk factor for major complications or death. Obesity was, however, shown to increase the risk for minor complications, namely perioperative surgical site infections [40]. In the past ten years, studies have been published to elucidate the impact obesity has on surgical outcomes in the setting of patients with PC undergoing CRS and HIPEC.

Votanopoulos et al. studied patients with PC originating from colorectal or appendiceal cancer who underwent CRS and HIPEC. Similarly, obesity was not a factor in predicting postoperative complications or death. Over the span of twenty-one years, a total of 925 CRS and HIPEC procedures were performed at this single institution, of which 272 of the procedures were performed on obese patients (BMI ≥ 30 kg/m^2^). Morbidity was found to be the same between obese and non-obese patients for minor and major complications presenting in the early postoperative (<30 days) and late postoperative (31–90 days) periods. Only re-admission in the late postoperative period was significantly higher for obese patients compared to non-obese (34.6% vs. 24.7%, *p* = 0.04). Additionally, when the obese patient cohort was further analyzed by their severity of obesity (moderately obese: BMI = 30 to 34.9 kg/m^2^ and severely obese: BMI ≥ 35 kg/m^2^), there were no noted differences in the length of surgery, occurrence of minor or major morbidity, length of hospital or intensive care unit stay, 30-day and 90-day mortality, and 30-day re-admission rates when each group was compared to non-obese patients. However, moderately obese patients were also noted to have higher rates of late postoperative re-admission than non-obese patients (35.8% vs. 24.7%, *p* = 0.05). Interestingly, this finding was not shared with the severely obese patient cohort (32.4% vs. 24.7%, *p* = 0.33). Obesity (BMI ≥ 30 kg/m^2^) was also found to not have clinical significance with overall survival when analyzed between the different primary tumor sites. However, severe obesity (BMI > 35 kg/m^2^) was found to have worse overall survival in low-grade appendiceal cancers when compared to non-obese patients (median overall survival: 54 vs. 107 months, *p* = 0.05). This finding was negated when the authors accounted for patient causes of death unrelated to the progression of the primary disease and was found to be related to underlying patient comorbidities. Obesity was concluded to not be a contributor to postoperative morbidity and mortality in this subset of patients undergoing CRS and HIPEC; thus, obesity should not be a contraindication to performing CRS and HIPEC in patients with PC [23].

In a 2019 study, Naffouje et al. also investigated the effect of obesity and BMI on surgical outcomes for patients with PC undergoing CRS and HIPEC. A total of 126 patients with PC from primary cancers including colorectal, appendiceal, ovarian, and pseudomyxoma peritonei, underwent CRS and HIPEC. Patients were subdivided into five groups based on BMI, including underweight, normal, overweight, obese, and morbidly obese. Overall, there was no difference in patient burden of comorbidities, preoperative albumin, peritoneal cancer index score, completion of cytoreduction, estimated blood loss, length of surgery, tumor grade/differentiation, and hospital length of stay across all BMI subgroups. Complications, as well as disease-free survival and overall survival, were also determined across each BMI group. When comparing across all BMI groups, there was no significant difference in postoperative complication rates for both minor and major complications (*p* = 0.231). Disease-free progression (underweight: 11.00 ± 0.85 months, normal: 15.00 ± 3.62 months, overweight: 35.00 ± 5.63 months, obese: 31.00 ± 8.88 months, morbidly obese: 29.00 ± 9.23 months, *p* = 0.035) and overall survival (underweight: 15.00 ± 2.57 months, normal: 27.00 ± 3.16 months, overweight: 66.00 ± 12.66 months, obese: 68.00 ± 27.74 months, morbidly obese: 48.00 ± 23.20 months, *p* = 0.001) distribution did significantly differ across the BMI spectrum with a trend towards longer disease-free survival and overall survival with increasing BMI. However, this protective factor of increasing BMI was no longer present at the extremes of obesity, as morbidly obese patients were found to have worse overall survival when compared to overweight and obese patients (*p* = 0.011) and in a multivariate regression analysis found to be an independent predictor of worse overall survival (HR: 1.823 [1.111–18.744], *p* = 0.043) [28].

On the opposite end of the spectrum, underweight BMI has also been associated with worse outcomes. Underweight BMI was found to be an independent predictor of worse disease-free survival and overall survival for all patients (HR: 45.826 [8.492–247.293], *p* < 0.001; HR: 21.583 [2.560–181.985]; *p* = 0.005) undergoing CRS and HIPEC [35]. This result is similar to the previously mentioned NSQIP study of patients undergoing major intra-abdominal cancer surgery, which showed that an underweight BMI was an independent predictor of postoperative mortality (OR: 5.24 [1.70–16.2], *p* = 0.0039) [33]. 

These studies and results underscore that the impact of BMI as a preoperative risk factor in patients with PC is not straightforward. The effect of BMI on surgical outcomes in PC patients varies with more extreme BMI values (i.e., underweight and morbidly obese) being associated with more complications and worse survival. These observations are an indication that BMI is not predictive of nutritional status and that many patients with increased BMI are likely malnourished and sarcopenic secondary to their malignancy. Obese patients, as well as underweight patients, should be thoroughly assessed in the preoperative setting for their nutritional status to critically evaluate their risks of postoperative morbidity and mortality. 

### 4.3. Clinical Impact of Sarcopenia/Cachexia

Sarcopenia and cachexia have consistently been implicated in negatively impacting postoperative gastrointestinal cancer surgery outcomes [7,41]. These studies, however, have not investigated sarcopenia and cachexia’s impact in the setting of PC treated with CRS and HIPEC. Recent interest has arisen in addressing the effects of sarcopenia and cachexia on outcomes for patients undergoing CRS and HIPEC. The reported effects of these clinical entities on outcomes in this patient population have varied. Some studies have shown a negative impact while others have shown no impact on outcomes.

van Vugt et al. conducted a study of 206 patients with colorectal primary cancer who had CRS and HIPEC. Patient sarcopenia, or skeletal muscle depletion, was determined through measuring mass muscle at the L3 vertebral level with skeletal muscle index values of less than 52.4 cm^2^/m^2^ for men and 38.5 cm^2^/m^2^ for women. The authors showed that sarcopenic patients had significantly lower BMI than nonsarcopenic patients (23.5 vs. 26.4 kg/m^2^), but all other preoperative factors were similar. Sarcopenic patients trended towards having more complications than non-sarcopenic patients (54.4% vs. 41.4%, *p* = 0.062), including severe complications (33.3% vs. 21.6%, *p* = 0.058), but these findings were not statistically significant. These patients, however, were statistically more likely to undergo reoperation for complications than non-sarcopenic patients (25.6 vs. 12.1%, *p* = 0.012). Higher L3 muscle mass was independently associated with less severe postoperative complications (OR: 0.93 [0.87–0.99], *p* = 0.018). The authors reported a similar 30-day mortality rate between the two groups: 2.2 vs. 2.6% for sarcopenic and non-sarcopenic patients, respectively, which was not significantly different [37]. The impact of sarcopenia on overall survival was not addressed in this study. 

Similarly, a 2016 study of 97 patients with varying primary cancers established a relationship between sarcopenia and perioperative complications. The authors explored sarcopenia’s relationship with complications attributed to the intraperitoneal chemotherapy and to the cytoreductive surgery independently. Sarcopenia was again defined by quantifying muscle mass on CT imaging. It should be noted that this study used a different definition for sarcopenia than the prior with sarcopenia representing a skeletal muscle index less than 41 cm^2^/m^2^ for women, less than 43 cm^2^/m^2^ for men with BMI ≤ 24.9 kg/m^2^, and less than 53 cm^2^/m^2^ for men with BMI >25 kg/m^2^. When analyzed individually, patients with sarcopenia were more likely to experience chemotherapy toxicities (57% vs. 26%, *p* = 0.004), namely chemotherapy-induced neutropenia (36% vs. 17%, *p* = 0.037), when compared to patients without sarcopenia. This relationship was confirmed in a multivariate analysis where sarcopenia was found to be the only significant independent predictor for chemotherapy toxicity (OR: 3.97 [1.52–10.39], *p* = 0.005). However, the authors found that sarcopenia was not associated with increased rates surgical complications (51% vs. 44%, *p* = 0.464) compared to no sarcopenia. In a multivariate analysis for predictors of postoperative complications, sarcopenia was not identified as a predictor of postoperative complications [38].

In a study of 115 patients with pseudomyxoma peritonei or mesothelioma carcinomatosis treated with CRS and HIPEC, preoperative sarcopenia was not predictive of postoperative morbidity. Sarcopenia was determined by measuring skeletal muscle mass on CT imaging at the L3 vertebral level and, again, the definition of sarcopenia differed compared to the aforementioned studies (skeletal muscle index ≤ 39 cm^2^/m^2^ for women and ≤55 cm^2^/m^2^ for men). Sarcopenic patients were significantly older (63.2 vs. 50.7 years, *p* = 0.002) and had lower BMI (22.3 vs. 24.2 kg/m^2^, *p* = 0.003) than non-sarcopenic patients. For all major hematologic, cardiovascular, respiratory, renal, and surgical complications, there was no difference in observed incidences between sarcopenic and non-sarcopenic patients. There was a trend toward higher incidence of death within 90 days of surgery in the sarcopenic group, though this did not achieve statistical significance (6.2% vs. 0%, *p* = 0.069) [39]. This study also assessed the impact of sarcopenia on overall survival. The authors showed that in all patients regardless of the primary cancer (pseudomyxoma peritonei or peritoneal mesothelioma), overall survival was significantly worse in sarcopenic patients compared to non-sarcopenic patients (57.2 vs. 73.3 months, *p* = 0.05) [39]. However, when examining overall survival by individual cancer type, median overall survival was worse in sarcopenic patients with pseudomyxoma peritonei (57.2 vs. 72.3 months, *p* = 0.045), but median overall survival was not different for sarcopenic versus non-sarcopenic in patients with peritoneal mesothelioma (39.2 vs. 57.7 months, *p* = 0.669). The most recently published study in 2020 reported similar findings. The definition of sarcopenia on CT imaging was defined as a skeletal muscle index of less than 52.4 cm^2^/m^2^ for men and 38.5 cm^2^/m^2^ for women, the same definition used by van Vugt et al. In a study population of 65 patients undergoing CRS and HIPEC for colorectal cancer-associated peritoneal carcinomatosis, overall survival was significantly decreased in sarcopenic patients compared to non-sarcopenic patients (17.7 vs. 37.9 months, *p* = 0.005). This relationship was further established on multivariate analysis where sarcopenia was reported to be a clinically significant predictor for increased mortality (HR: 2.246 [0.996–5.067], *p* = 0.050) [40]. The results of this and the prior study emphasize that overall survival is greatly impacted by the primary cancer type, arguing that the grouping of multiple cancer primaries into a single survival analysis to explore the impact of sarcopenia should be avoided and interpretation of such results should be done so cautiously.

On the contrary, sarcopenia has also been shown to not be associated with overall survival. In this study of 214 patients with primary colorectal cancer, skeletal muscle indices of less than 52.4 cm^2^/m^2^ for men and 38.5 cm^2^/m^2^ for women were used as thresholds for determining sarcopenia. Banaste et al. showed that sarcopenia did not impact overall survival (59 vs. 50 months, *p* = 0.648) or progression free survival (14 vs. 12 months, *p* = 0.364). Instead, preoperative hypoalbuminemia (OR: 0.562 [0.184, 0.949], *p* = 0.037) and occurrence of a major complication (OR: 0.465 [0.258, 0.838], *p* = 0.011) were significant predictors for overall survival. Also, when comparing patients who did or did not experience a major complication after CRS and HIPEC, there was no difference in the presence of sarcopenia in these two groups (42.5% vs. 42.0% for patients with no complication compared to patients with a major complication, respectively, *p* = 0.907). The authors concluded that preoperative imaging assessment for sarcopenia is not beneficial in predicting postoperative morbidity and mortality in this cohort [26].

As addressed in the preceding section, the clinical impact of the depletion of skeletal muscle mass in patients with PC managed with CRS and HIPEC is inconsistent across the currently available studies. It has been shown to influence postoperative outcomes both negatively or not at all. 

## 5. Conclusions

There is a paucity of data investigating the impact of BMI, sarcopenia, and cachexia on patients with PC treated with CRS and HIPEC, with only seven studies currently published and presenting inconsistent results. All the aforementioned studies focused on carcinomatosis of varying primary origin with some studies including multiple cancer types. Patient survival and morbidity is dependent upon the primary tumor as certain types of carcinomatosis respond better to CRS and HIPEC; thus, it is difficult to generalize results when multiple cancer primaries are grouped together in studies. The definition of sarcopenia defined by quantification of skeletal muscle mass also differed among all of the studies, which further complicates the interpretation of the results. Furthermore, five of the seven studies were conducted in Europe (France, Netherlands, and Turkey) and the remaining two studies were from the United States. There is no reporting of race or ethnicity in these studies which raises the question of how generalizable these results can be with different patient populations wherever CRS and HIPEC is performed throughout the world. Cachexia has been shown, for instance, in pancreatic cancer to afflict different races and ethnicities disproportionately [42]. Further studies that investigate the impact of cachexia on patients undergoing CRS and HIPEC in the context of primary cancer type, race, and other factors will provide a better understanding of this disease process. In conclusion, a better appreciation of how cachexia impacts clinical outcomes in patients with peritoneal carcinomatosis will provide a better understanding of this disease process and improve overall survival. 

## Figures and Tables

**Figure 1 cancers-14-02853-f001:**
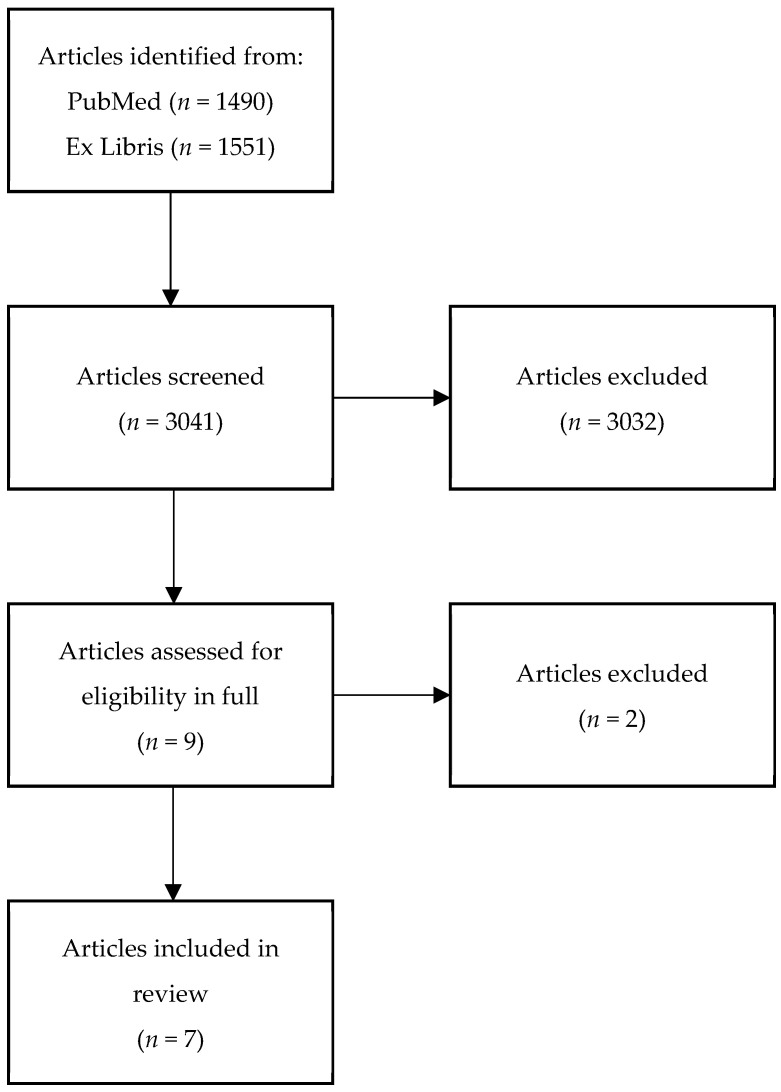
PRISMA diagram of the literature review.

**Table 1 cancers-14-02853-t001:** Summary of key findings from studies included in review.

Study	Location of Study	Years of Study	No. of Patients	Primary Cancer	Surgical Therapy	Key Findings
Votanopoulos et al., 2013 [23]	USA	1991–2012	246	Colorectal Appendiceal	CRS/HIPEC	Obesity is not a predictor of postoperative complications or death
van Vugt et al., 2015 [24]	Netherlands	2005–2013	206	Colorectal	CRS/HIPEC	Sarcopenic patients are more likely to undergo re-operation for complications Decreased L3 muscle mass is independently associated with a higher risk of severe postoperative complications
Chemama et al., 2016 [25]	France	2008–2010	97	Colorectal Appendiceal Mesothelioma Other (Unspecified)	CRS/HIPEC	Sarcopenia is independently associated with increased risk of experiencing chemotherapy toxicities (e.g., neutropenia) Sarcopenia is not associated with risk of postoperative complications from cytoreductive surgery
Banaste et al., 2018 [26]	France	2009–2014	214	Colorectal	CRS ± HIPEC	Preoperative hypoalbuminemia is independently associated with worse overall survival Sarcopenia is not associated with overall survival
Galan et al., 2018 [27]	France	2009–2017	115	Pseudomyxoma peritonei Mesothelioma	CRS/HIPEC	Preoperative sarcopenia is not predictive of postoperative morbidity Preoperative sarcopenia is a predictor of overall survival
Naffouje et al., 2019 [28]	USA	2007–2017	126	Colorectal Appendiceal Ovarian Pseudomyxoma peritonei	CRS/HIPEC	Underweight BMI is independently associated with poor prognosis for disease free survival and overall survival Morbidly obese BMI is independently associated with poor prognosis for overall survival
Agalar et al., 2020 [29]	Turkey	2008–2018	65	Colorectal	CRS/HIPEC	Preoperative sarcopenia is associated with increased risk of morbidity and mortality Overall survival is decreased in patients with sarcopenia

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
