# Peer review of "A Review of the Clinical Implications of Cachexia, Sarcopenia, and BMI in Patients with Peritoneal Carcinomatosis Receiving Cytoreductive Surgery and Hyperthermic Intraperitoneal Chemotherapy"

_cancers, 2022, doi:10.3390/cancers14122853_

Round 1

Reviewer 1 Report

The authors conducted a comprehensive systemic review of current literatures regarding to the impact of cachexia/sarcopenia and obesity/BMI on patients with peritoneal carcinomatosis receiving cytoreduction surgery followed by hyperthermic intraperitoneal chemotherapy. This review is clinically useful for physicians who treated patients with peritoneal carcinomatosis. However, there existed some questions warranted further explanations.

1.     The authors review the current literature investigating the impact of  sarcopenia, cachexia, and body mass index on outcomes in this patient population that undergo surgical treatment. However, the title of study was “The Clinical Implications of Cytoreductive Surgery and Hyperthermic Intraperitoneal Chemotherapy on Peritoneal Carcinomatosis-induced Cachexia”. This really confused me. Whether the authors want to investigate the impact of cachexia on survival or the authors want to discuss the effects of CRS/HIPEC in the treatment of cachexia?  In my opinion, the title should be revised to “ The Clinical Implications of Cachexia in Patients with Peritoneal Carcinomatosis receiving Cytoreductive Surgery and Hyperthermic Intraperitoneal Chemotherapy. “ 

2.     The authors investigated the impact of cachexia, sarcopenia, BMI. However, the title only mentioned about cachexia. Hence, the authors need to introduce the relationship between cachexia, sarcopenia and BMI. Otherwise, the authors should make a more precise title according to the content of manuscript. 

3.     The authors demonstrate that BMI was found to not be a contraindication to surgery, but the impact of BMI on survival and post-surgical complication was variable. As we know, BMI can be classified into underweight, normal , overweight and obese. Thus, the authors should separately investigate the impact of different range of BMI on survival and post-surgical complication. Dose the impact remain variable across all range of BMI? The authors might try to make the conclusion according to different BMI and give the possible explanations about their conclusions. 

4.     The authors reported that they adopted the most recent 2018 consensus EWGSOP in their literature search to ensure all relevant studies were identified. However, this review enrolled many studies published before 2018. With the inconsistent definition of sarcopenia, the impact of sarcopenia on those patients receiving CRS and HIPECT should also be evaluated separately according to the definition.

5.     The authors finally concluded that the evaluation about the impact of BMI, sarcopenia, and cachexia on patients with PC undergoing CRS and HIPEC was difficult because of limited data. There are also inconclusive results based on this study. So, this review might be re-conducted several years later until more data available to draw a more conclusive result.

Author Response

The authors conducted a comprehensive systemic review of current literatures regarding to the impact of cachexia/sarcopenia and obesity/BMI on patients with peritoneal carcinomatosis receiving cytoreduction surgery followed by hyperthermic intraperitoneal chemotherapy. This review is clinically useful for physicians who treated patients with peritoneal carcinomatosis. However, there existed some questions warranted further explanations.

  1. The authors review the current literature investigating the impact of sarcopenia, cachexia, and body mass index on outcomes in this patient population that undergo surgical treatment. However, the title of study was “The Clinical Implications of Cytoreductive Surgery and Hyperthermic Intraperitoneal Chemotherapy on Peritoneal Carcinomatosis-induced Cachexia”. This really confused me. Whether the authors want to investigate the impact of cachexia on survival or the authors want to discuss the effects of CRS/HIPEC in the treatment of cachexia?  In my opinion, the title should be revised to “The Clinical Implications of Cachexia in Patients with Peritoneal Carcinomatosis receiving Cytoreductive Surgery and Hyperthermic Intraperitoneal Chemotherapy. “ 

Reviewer #1, thank you for your comments and feedback and pointing out that the wording of the title was confusing.  We agree that the title was confusing and did not accurately reflect the paper.  We have revised the title as you suggested and made it more comprehensive: “A Review of The Clinical Implications of Cachexia, Sarcopenia, and BMI in Patients with Peritoneal Carcinomatosis Receiving Cytoreductive Surgery and Hyperthermic Intraperitoneal Chemotherapy.”

  1. The authors investigated the impact of cachexia, sarcopenia, BMI. However, the title only mentioned about cachexia. Hence, the authors need to introduce the relationship between cachexia, sarcopenia and BMI. Otherwise, the authors should make a more precise title according to the content of manuscript. 

Reviewer #1, we agree that the title was not comprehensive enough.  We had edited it to include sarcopenia and BMI, as well as adding the word “review” to ensure readers know that this paper is a review and not an original investigation.

  1. The authors demonstrate that BMI was found to not be a contraindication to surgery, but the impact of BMI on survival and post-surgical complication was variable. As we know, BMI can be classified into underweight, normal , overweight and obese. Thus, the authors should separately investigate the impact of different range of BMI on survival and post-surgical complication. Dose the impact remain variable across all range of BMI? The authors might try to make the conclusion according to different BMI and give the possible explanations about their conclusions. 

Reviewer #1, thank you for addressing that outcomes can vary based on BMI.  We have addressed this in the main body of the review.  In summary, obesity was found to not be a contraindication to surgery.  Two studies also found that BMIs at the extremes of the spectrum (underweight vs morbidly obese) had worse survival outcomes.  This is now discussed extensively in the text. 

  1. The authors reported that they adopted the most recent 2018 consensus EWGSOP in their literature search to ensure all relevant studies were identified. However, this review enrolled many studies published before 2018. With the inconsistent definition of sarcopenia, the impact of sarcopenia on those patients receiving CRS and HIPEC should also be evaluated separately according to the definition.

To address the varying definitions of sarcopenia, we have specifically cited what definition each study used when determining whether a patient was sarcopenic or not.  These varied across the studies and there was no universal definition. We have addressed how this may also impact the generalizability of the results.

  1. The authors finally concluded that the evaluation about the impact of BMI, sarcopenia, and cachexia on patients with PC undergoing CRS and HIPEC was difficult because of limited data. There are also inconclusive results based on this study. So, this review might be re-conducted several years later until more data available to draw a more conclusive result.

Reviewer #1, thank you for the feedback.  We believe that despite the limited number of current studies and inconsistent results across the studies, that this review still merits publication at this time.  This is because it shows that this field needs more researchers to actively participate in with larger patient sample sizes of each possible cancer type so that the impact of these entities can be better understood in this patient population.

Reviewer 2 Report

I suggest several modifications prior the acceptance for publication:

-To change the title: to add an information that is a review study.

-Abstract: How many studies were included? Why did not include values? Number of studies assessed?

-Introduction: What is the aim of this review?

-Methods: This is a narrative or descriptive review? 

Major revisions:

What is the definition of cachexia? In addition, the authors concluded that cachexia impacts clinical outcomes in patients with peritoneal carcinomatosis. But, the sarcopenia there is an important impact on prognosis. 

Author Response

I suggest several modifications prior the acceptance for publication:

-To change the title: to add an information that is a review study.

Reviewer #2, thank you for your comments and feedback in strengthening this paper.  We had edited the title to make it clearer and it more comprehensive, including identifying that this is a review paper.  The title now reads “A Review of The Clinical Implications of Cachexia, Sarcopenia, and BMI in Patients with Peritoneal Carcinomatosis Receiving Cytoreductive Surgery and Hyperthermic Intraperitoneal Chemotherapy.”

-Abstract: How many studies were included? Why did not include values? Number of studies assessed?

Reviewer #2, thank you for your comment.  We have now included this in the abstract: “In total 3,041 articles were screened and only seven original studies met inclusion criteria for review.”

-Introduction: What is the aim of this review?

Reviewer #2, thank you for the comment.  We have now clearly stated in the introduction the aim of this review: “The aim of this article is to review the current literature regarding the mechanisms of cachexia in the PC patients with CRS and HIPEC, and the effects of skeletal muscle loss or sarcopenia, cachexia, and BMI on patient outcomes.”

-Methods: This is a narrative or descriptive review? 

Reviewer #2, we consider this to be a narrative review.

Major revisions:

What is the definition of cachexia? In addition, the authors concluded that cachexia impacts clinical outcomes in patients with peritoneal carcinomatosis. But, the sarcopenia there is an important impact on prognosis. 

Reviewer #2, thank you for addressing this concern. The definition is now clearly addressed in the main text. With regards to sarcopenia, we have specifically cited which definition each study used when determining whether a patient was sarcopenic or not. These varied across studies and we addressed how this may also impact the generalizability of the results.

Round 2

Reviewer 1 Report

The authors have revised their manuscript according to our suggestions. I think this paper is adequate for publication now.

Reviewer 2 Report

The paper has improved. I suggest the acceptance.